# Association between Circulating MicroRNAs (miR-21-5p, miR-20a-5p, miR-29b-3p, miR-126-3p and miR-101-3p) and Chronic Allograft Dysfunction in Renal Transplant Recipients

**DOI:** 10.3390/ijms232012253

**Published:** 2022-10-14

**Authors:** Yu-Jen Chen, Chia-Tien Hsu, Shang-Feng Tsai, Cheng-Hsu Chen

**Affiliations:** 1Division of Nephrology, Department of Internal Medicine, Taichung Veterans General Hospital, Taichung 407219, Taiwan; 2Department of Life Science, Tunghai University, Taichung 407224, Taiwan; 3Department of Post-Baccalaureate Medicine, College of Medicine, National Chung Hsing University, Taichung 40227, Taiwan; 4School of Medicine, China Medical University, Taichung 651012, Taiwan

**Keywords:** chronic allograft dysfunction, microRNA, biomarker

## Abstract

Chronic allograft dysfunction (CAD) is a major condition affecting long-term kidney graft survival. Serum microRNA (miRNA) has been reported as a biomarker for various conditions of allograft injuries. The upregulation of miR-21 is the best-known miRNA change in graft tissue, urine and plasma. However, the correlation of plasma miR-21 with the severity of CAD remains unclear. In our study, 40 kidney transplantation recipients with late graft survival for more than 10 years were enrolled. The CAD group (*n* = 20) had either an eGFR between 15 to 60 mL/min or a biopsy-proved chronic allograft nephropathy or rejection. The control group (*n* = 20) had an eGFR ≥ 60 mL/min without proteinuria and hematuria for a consecutive 3 months before the study. We performed RNA sequencing to profile the miRNAs expression. There were six differentially expressed miRNAs in the CAD group. Among them, miR-21-5p and miR-101-3p were decreased, and miR-20a-5p was increased. We found that miR-21-5p, miR-20a-5p and miR-101-3p all participated in the TGF-beta pathway. We demonstrated that decreased miR-21-5p and miR-101-3p, and increased miR-20a-5p were the novel CAD-associated miRNAs in the TGF-beta pathway. These findings may pave the way for developing early prediction miRNAs biomarkers for CAD, and possibly developing therapeutic tools in the field of kidney transplantation.

## 1. Introduction

Potent immunosuppressive agents markedly improve renal graft survival rates. The 5-year graft survival rate of deceased transplant donors (DTD) combined with living transplant donors (LTD) was 88.7% in Taiwan during the years 2009 to 2013 [1]. Meanwhile, in the USA during the period 2012 to 2015, the graft survival rate of LTD was 93.29% and that of DTD was 78.15% [2]. Maintaining long-term graft survival remains a major unmet need. The 10-year patient graft survival rate of deceased kidney transplant recipients was 53.62%, and that of living kidney transplants was 81.28% in the USA from 2008 to 2011 [2]. The leading etiology of kidney graft failure includes alloimmune injury and recurrent glomerulonephritis [2]. Although a kidney biopsy can provide valuable information on the differential diagnosis of allograft dysfunction, it is an invasive procedure and is not always readily available. Chronic allograft dysfunction (CAD) is defined as a clinical condition characterized by a slowly progressive drop in kidney function, usually associated with hypertension and proteinuria. The etiology of CAD includes rejection, BK polyomavirus nephropathy, graft renal artery stenosis, and calcineurin inhibitor (CNI) nephropathy. MicroRNAs (miRNAs) are small non-coding RNAs with lengths ranging from 18 to 24 nucleotides. They are stable in tissues and body fluids, and are promising, non-invasive biomarkers for disease diagnosis and forensic science [3,4]. Additionally, they are preserved in eukaryotic organisms and regulate the post-transcriptional expression of genes, while providing insight into the molecular pathway of different renal injuries following kidney transplantation. Previous kidney allograft studies have focused on acute kidney injury, acute or chronic rejection and interstitial fibrosis/tubular atrophy (IF/TA). Kidney fibrosis is observed in the graft histology of CAD patients, and the TGF-β pathway is well known for causing tissue fibrosis. Several microRNAs have been reported to regulate the TGF-β pathway in animal models and human studies [5]. In samples of graft biopsies revealing IF/TA, their miR-21 levels had increased [6,7]. Additional studies have also suggested that both plasma miR-21 [7,8,9] and urinary miR-21 [10,11] increased in IF/TA. Other miRNAs, including miR-142-3p, miR-142-5p, miR-155, miR-200b and miR-29 are also related to IF/TA [6,9,12]. Here, we aimed to identify miRNAs signatures in long-term CAD and also explore any potential targets for diagnosis and treatment.

## 2. Results

Demographic data surrounding the two groups are shown in Table 1. No inter-group differences were found regarding age, graft survival, donor status or malignancy after transplantation. There were more females in the CAD group compared with the control group (12 vs. 5, *p* = 0.025), while the FK-506 level was higher in the CAD group (5.7 ± 0.9 vs. 5.1 ± 1.2, *p* = 0.829). Additionally, there were apparently fewer cases of malignancy in the CAD group when compared with the control group (2 vs. 5, *p* = 0.407).

In the CAD group, the histology of renal specimens revealed five mildly acute T-cell rejections, two chronic antibody-mediated rejections (CAMRs), two tubular injuries and one striped fibrosis. In the control group, two patients had undergone a biopsy 10 years after kidney transplantation, and both of their histology revealed acute tubular injury.

Of the eighty-nine miRNAs in the CAD group, we found eight to be differentially expressed (i.e., miR-21-5p, miR-15a-5, miR-101-3p, miR-589-5p, miR-122-5p, miR-20a-5p, miR-29b-3p and miR-126-3p). Amongst them, three were up-regulated (miR-20a-5p, miR-589-5p and miR-29b-3p) and five were down-regulated (miR-21-5p, miR-15a-5, miR-101-3p, miR-122-5p and miR-126-3p) in the CAD group. The miRNAs were identified in accordance with our criteria of fold change (≥±0.585) and *p* values (<0.05) as shown in Table 2 and Figure 1.

The clustering analysis and PCA plot can be found in Appendix A. In silico analysis on the targeted genes of the eight differentially expressed miRNAs was performed, and we found miRNA target interaction (MTI) in miRTarBase [14]. The top ten GO enrichments of the biological process (BP) are listed in Table 3. Extracellular matrix organization, extracellular structure organization and external encapsulating structure organization were three of the top four terms according to their *p*-value. Other GO terms are listed in Appendix A. 

## 3. Discussion

In this case-control study, we explored the miRNAs signature of CAD with a graft survival period of more than 10 years. Of the eight differentially expressed miRNAs, miR-21-5p, miR-15a-5p, miR-101-3p, miR-122-5p and miR-126-3p were detected in more than nineteen samples in both the CAD and control groups. Enrichment analyses revealed that miR-21-5p, miR-20a-5p, miR-101-3p, miR-126-3p and miR-29b-3p participated in the TGF-B/Smad pathway.

miR-21 plays a crucial role in development, cancer, cardiovascular kidney disease, the aging process and inflammation [15]. Ghorbanmehr et al. found that urinary miR-21-5p could be differentially detected in prostate and bladder cancer patients [16]. A growing body of literature suggests that miR-21 plays an important role in chronic kidney disease in animal models and human studies [17,18]. miR-21 was upregulated in the TGF-β1/Smad pathway, as reported in several studies [19,20]. TGF-β is known for its anti-inflammatory, anti-neoplasm and fibrosis functions. The TGF-β canonical pathway is illustrated and described in Figure 2, however the non-canonical pathway was beyond the scope of our study. The association between miR-21 and fibrosis can be explained by its effect on the down-regulation of Smad7, which inhibits Smad3 [19,20]. Zang et al. found elevated urinary exosome mir-21-5p in diabetic kidney disease patients [21]. Elevated levels of miR-21 in human graft biopsy tissue were found to be associated with tubulointerstitial fibrosis [6], while elevated urinary miR-21 has also been associated with elevated levels of miR-21 in graft tissue [10,11,12]. However, a similar correlation with plasma miR-21 levels is less clear. Glowacki et al. reported that plasma miR-21 elevation is associated with severe IF/TA in graft kidney [7]. Alternatively, Saejong et al., reported that whole-plasma miR-21 is lower in CAD cases, a finding that is consistent with our present results. Interestingly, the plasma exosome miR-21 level was higher in CAD, thus making it better correlated with the severity of IF/TA [8].

miR-126-3p was reported to be associated with malignancy [22,23,24], particularly in lung and breast cancer patients [25,26,27,28,29]. Jordan NP et al. found that miR-126-3p was down-regulated in the fibrotic tissue of murine kidney and heart, with the process being due to the endothelial-to-mesenchymal transition [30]. Motshwari et al. found increased whole-blood miR-126-3p in a South African community-based sample of subjects diagnosed with chronic kidney disease as compared to a control group [31]. Although there was no kidney transplant study performed, Schaefer et al. demonstrated that pre-treatment with S-NO-HSA led to reduced fibrosis and the preservation of myocardial miR-126-3p and GATA2 levels in murine cardiac isografts 60 days after transplantation [32]. Therefore, miR-126-3p may serve as a potential therapeutic target in the organ transplantation field.

miR-101-3p has been reported on over the last 3 years. Wang et al. found miR-101 reverse TGFβ1 induced epithelial-to-mesenchymal transition (EMT) in HK 2 cells with less fibrosis [33]. Zhao et al. also revealed that miR-101 inhibits AKI-to-CKD transition by regulating EMT [34]. One additional study also demonstrated that miR-101 suppresses chronic renal fibrosis by regulating KDM3A, hence blocking the YAP-TGF-β/Smad signaling pathway [35]. Interestingly, one study showed that dexmedetomidine potentially protects against renal fibrosis by targeting the miR-101/TGF-β/Smad pathway. miR-101-3p may serve as a promising diagnostic marker of CAD. However, further studies surrounding histology, urinary and plasma expression in patients with kidney transplants are still required.

miR-20a-5p can repress myoblast proliferation in avian cells [36]. It was also reported to be upregulated during the first few days after acute ischemic reperfusion injury in a mouse model [37]. Its correlation with fibrosis in kidney disease was not made clear, but miR-20a-5p was down-regulated in liver fibrotic tissue [38]. Smad4 was the target of miR-20 in the pathway [39]. Interestingly, several studies have suggested that miR-20a-5p could be a biomarker for malignancy due to Smad4 serving as a co-Smad in the TGF-β pathway [40,41,42,43]. We also observed fewer malignancy cases in the CAD group. In the kidney disease studies, Smad4 deficiency in a unilateral ureteral obstruction (UUO) model likely enhances both renal inflammation and fibrotic response [20]. To the best of our knowledge, the association between miR-20 and human chronic kidney disease or allograft injury has never been reported.

**Figure 2 ijms-23-12253-f002:**
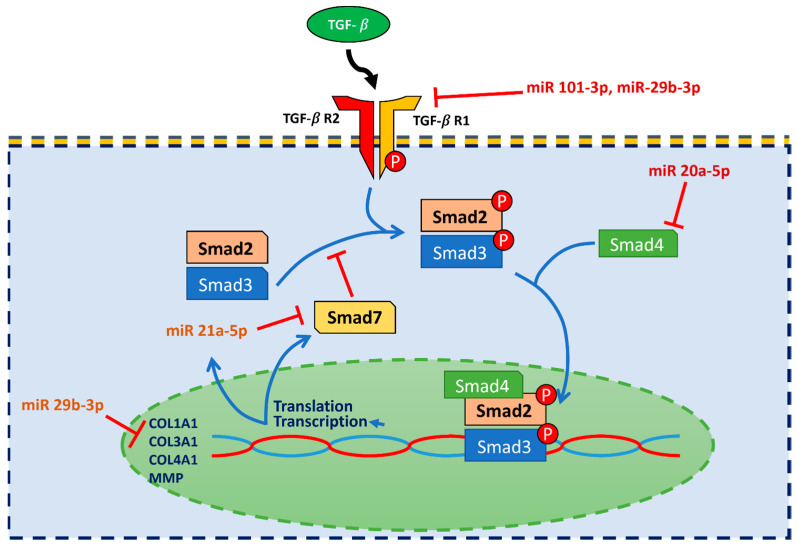
The TGF-β/Smad canonical pathway is illustrated. TGF-β could activate TGFR1 and TGFR2, inducing phosphorylation of Smad2 and Smad3. Phosphorylated Smad2/Smad3 enters the nucleus with Smad4 and promotes pro-fibrotic gene expression. One of the consequences of the TGF-β/Smad canonical pathway gene expression was upregulation of miR-21-5p. MiR-21-5p inhibited Smad7, thus promoting fibrosis via disinhibition of Smad2 and Smad3 phosphorylation [19]. Other miRNAs could target different molecules in the TGF-β/Smad pathway. MiR-20a-5p targeted Smad4 [40]. MiR-101-3p and miR-29b-3p [5] could inhibit TGF-β/Smad through TGFR1 inhibition [35]. MiR-29b-3p could also inhibit the collagen expression gene [5]. The combined effect of different miRNAs could possibly enhance the fibrogenesis process.

The miR-29 family has been well known for their inhibition of extracellular matrix-related genes, including COL1A1, COL3A1, COL4A1, MMP2, TGFB1 and TGFB2 [5,44,45,46,47] for more than 10 years, while also having the most potential as a therapeutic target for antifibrosis treatment [48]. Zhang et al. found that long noncoding RNA Tug1 promotes angiotensin-II-induced renal fibrosis by interacting with miR-29b-3p [49]. Wang et al. demonstrated that exosome-mediated miR-29 transfer could reduce muscle atrophy and kidney fibrosis UUO in a mice model [50]. Zhang et al. also found that rAAV9-mediated supplementation could improve angiotensin-II-induced renal fibrosis in mice [51]. Gondaliya et al. also demonstrated that miR-29b could attenuate histone deacetylase-4-mediated podocyte dysfunction and renal fibrosis in diabetic nephropathy [52]. Although miR-29 is upregulated in fibrotic tissue, serum level has not frequently been reported to be correlated with fibrosis. Rubis et al. found decreased miR-21-5p and elevated miR-29b in biopsy-proven fibrotic dilated cardiomyopathy patients [53]. Therefore, miR-29b-3p could be a promising therapeutic target for allograft dysfunction.

As for miR-15a-5p and miR-122-5p, several studies have been published regarding fibrosis in different organs [54,55,56,57]. Liu et al. compared different treatments involving rAAV-miR-122-5p and rAAV-GFP in 14-week-old male SHR and WKY rats. The team demonstrated that miR-122-5p possesses potential therapeutic significance for hypertensive renal injury and fibrosis-related kidney diseases [58]. Dieter et al. examined the urine and plasma levels of miR-15a-5p and miR-30e-5p in patients with type I DM and CKD., with no difference being found regarding miR-15a-5p levels [59]. The above two miRNAs still require more studies to be performed in order to confirm their roles in human kidney disease.

Multiple pieces of evidence surrounding elevated miR-21 being in association with renal fibrosis have been reported. However, in our patients, we observed a lower plasma miR-21-5p level compared with the control group. Li et al. found elevated circulating miR-21-5p after steroid treatment had been performed in humans and rats [60], possibly indicating that anti-inflammatory status may be correlated with elevated circulating miR-21-5p. We postulated that miR-21-5p, miR-20a-5p and miR-29b-3p serve as a housekeeping function, are secreted by various cells and can be detected at a baseline level. When allograft dysfunction gradually developed, allograft kidney increased the uptake of miR-21-5p and decreased the uptake of miR-29b-3p and miR-20a-5p. A serial follow up of multiple microRNAs could be utilized as a surveillance or prediction tool for allograft dysfunction. Hromadnikova et al. were able to predict that 47.93% of patients were destined to develop gestational diabetes mellitus (GDM) at a 10.0% false-positive rate during the first trimester of pregnancy, with 11 dysregulated microRNAs. After incorporating clinical characteristics, the team increased their detection rate to 72.5%, with a 10% false-positive rate [61]. We will consider combining the microRNAs together and then evaluating their performance when predicting CAD.

Our study had the limitation of potential bias due to its retrospective nature and small sample size. Additionally, we did not implement the protocol-directed biopsy in our hospital. Consequently, a heterogeneity of diagnosis likely existed in the allograft dysfunction group, and therefore the interpretation of the results should be prudent.

In conclusion, the plasma microRNA signature of downregulated miR-21-5p, miR-101-3p and miR-126-3p, as well as upregulated miR-20a-5p and miR-29b-3p is a promising biomarker for detecting CAD. MiR-29b-3p offers great potential towards attenuating chronic allograft dysfunction.

## 4. Materials and Methods

### 4.1. MicroRNA Extraction

Blood samples were collected, and plasma was extracted within 2 h of admittance according to standard procedures, with the samples first being centrifuged at 1200× *g* for 10 min, and the supernatant again being centrifuged at 12,000× *g* for 10 min. The supernatant, which was plasma, was divided into aliquots and immediately stored at −80 °C. The miRNA was isolated from 200 μL of plasma according to the miRNeasy manufacturer’s protocol. Plasma miRNAs were eluted in 20 μL of nuclease-free water. Concentrations of the extracted miRNAs were quantified using the Qubit microRNA Assay Kit (Q32880, ThermoFisher Scientific Inc., Waltham, MA, USA).

### 4.2. cDNA Synthesis & qPCR Analysis

From the samples, 2 ng of the total miRNAs were used to synthesize cDNA with 20 μL reverse transcription reactions. The reverse transcription step was performed as follows: poly-A tail was added to the miRNA population using Poly-A polymerase, followed by cDNA synthesis with a QuarkBio’s microRNA Reverse Transcription Kit (Quark Biotechnology, Inc., Zhubei, Taiwan). The qPCR was performed utilizing the NextAmp™ Analysis System and mirSCAN™ V2 PanelChip^®^. For qPCR analysis, 0.15 ng cDNA was added to the QuarkBio qPCR master mix (Quark Biotechnology, Inc., Zhubei, Taiwan), and Q Station™ (Quark Biotechnology, Inc., Zhubei, Taiwan) was run according to the following cycle program: 95 °C for 36 s and 60 °C for 72 s over 40 cycles. For miRNA expression profile analysis, SYBR Green-based qPCR methodology used the miScriptmiRNA PCR Array Human T-Cell and B-Cell Activation panel (Qiagen, Las Matas, MD, Spain), allowing for the quantification of 89 miRNAs. This method was used to identify those miRNAs differentially expressed (DE) in 20 transplant samples with CAD and 20 transplant samples having stable renal function. RT-PCR data were normalized according to the manufacturer’s instructions. The miRNAs were selected based on the fold changes of ≥±0.585 and at *p* values < 0.05.

### 4.3. The miRNA-Target Functional Analysis

Once the differentially expressed miRNAs were detected, we determined miRNA target interaction (MTI) using miRTarBase. We filtered out the MTIs with less than 3 reference supports as well as any non-functional MTIs. Genes listed within MTIs were then analyzed for gene set enrichment using clusterProfiler. Gene Ontology, KEGG Pathway and Disease Ontology were used for functional analysis due to their long-standing curation.

### 4.4. Pathway and Gene Ontology Analysis

We performed gene set enrichment analysis on Pathway terms and Gene Ontology (GO) terms with the input of target genes from differentially expressed miRNAs. Overrepresented gene sets were performed using GOSemSim in order to calculate the similarity of GO terms and remove the highly similar terms by keeping one representative term. A background gene set was based on validated miRNA target interactions indicated by the miRTarbase (2766 genes).

### 4.5. PCA Analysis

Principal component analysis (PCA) was performed in order to evaluate the differences between biological replicates and their treatment conditions. PCA adopted an orthogonal transformation to convert a set of observations of possibly correlated variables into a set of uncorrelated variables called principal components.

### 4.6. Clustering Analysis

For advanced data analyses, intensity data were pooled and calculated to identify differentially expressed microRNAs based on the threshold of fold changes and *p*-values. The correlation of expression profiles between samples and treatment conditions was demonstrated through unsupervised hierarchical clustering analysis.

### 4.7. Statistics

Continuous variables were summarized as mean ± standard error or median with interquartile range. Continuous variables with normal distribution were compared using Student’s *t*-test or Fisher’s exact test, and those with non-normal distribution were compared with the Mann–Whitney U test. Categorical variables were compared using the χ2 test or Fisher’s exact test. All statistical tests were 2-tailed, and a *p* value < 0.05 was considered statistically significant. Statistical analyses were conducted using IBM SPSS Statistics 26 (SPSS Inc., Chicago, IL, USA). Plots were generated using Prism 6 (GraphPad, San Diego, CA, USA).

### 4.8. Patients

We retrospectively recruited a total of 40 kidney transplant recipients who were experiencing late renal graft survival for a period of more than 10 years from Taichung Veteran General Hospital. We recorded their age, gender, graft survival, donor status, cause of end stage renal disease and malignancy after transplantation. All blood samples were collected after receiving their informed consent. Patient data included estimated glomerular filtration fraction (eGFR), plasma creatinine concentration (SCr), urine protein creatinine ratio, calcineurin inhibitor dosage, calcineurin inhibitor plasma level, prednisolone daily dosage and mycophenolic acid daily dosage. The eGFR was calculated from plasma creatine using the Modification of Diet in Renal Disease (MDRD) equation.

Recipients in the CAD group (*n* = 20) included patients with their eGFR between 15 and 60 mL/min, or those who had biopsy-proved chronic transplant allograft nephropathy or rejection. The control recipients (*n* = 20) had an eGFR ≥ 60 mL/min without either proteinuria or hematuria for 3 consecutive months prior to the study. We performed cDNA synthesis and RT-PCR on the plasma of each patient in order to profile miRNA expressions and predict the potential targets of differentially expressed miRNAs in CAD. There were 10 patients in the CAD group and 2 patients in the control group who had previously received a kidney allograft biopsy prior to blood sampling.

## Figures and Tables

**Figure 1 ijms-23-12253-f001:**
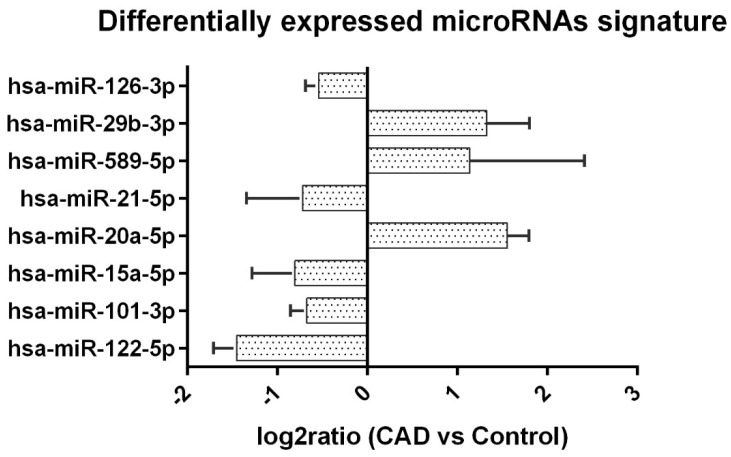
The differentially expressed microRNA in fold change (log2ratio). The eight microRNAs were differentially expressed. The fold change (log2ratio) of Cq level is shown with bar and standard deviation.

**Table 1 ijms-23-12253-t001:** Baseline characteristics.

Variable	CAD *n* = 20	Control *n* = 20	*p*-Value
Age (year)	53 (10.7)	55.3 (8.3)	0.399
Gender (Male), %	8 (40%)	15 (75%)	0.025
Donor, %			1.000
Cadaveric KT	15 (75%)	16 (80%)	
Living KT	5 (25%)	4 (20%)	
Graft Survival (year)	17.5 ± 4.4	17 ± 3.8	0.754
eGFR (ml/min)	43.9 ± 9.1	74.4 ± 7.3	0.0
Creatinine (mg/dL)	1.6 ± 0.3	1 ± 0.1	0.0
UPCR (unit)	685.8 ± 1646.7	127.2 ± 75	0.161
Tacrolimus (mg/day)	5.7 ± 2.9	3.6 ± 1.8	0.026
FK-506 level (mg/dL)	5.7 ± 0.9	5.1 ± 1.2	0.829
Cyclosporin (mg/day)	(*n* = 1, 100 mg/day)	150 ± 35.35	0.206 *
CSA level (mg/dL)	(*n* = 1, 77.4 mg/dL)	84.6 ± 23.69	0.480 *
Prednisolone (mg/day)	5.6 ± 3.4	3.9 ± 2.2	0.150
Malignancy (Positive), %	2 (10%)	5 (25%)	0.407
ESRD causes			
IgA nephropathy	1	4	<0.05
Herb nephropathy	0	3	NS
FSGS	3	1	NS
ADPKD	0	3	NS
Lupus Nephropathy	3	0	NS
NSAID Nephropathy	1	1	NS
Hypertensive Nephropathy	2	0	NS
Undiagnosed GN	9	8	NS
MPGN	2	0	NS
DM	1	0	NS
VUR	1	0	NS

The baseline characteristics of allograft dysfunction and the control group. The dysfunction group had more females and higher FK-506 levels (*p* < 0.05), KT: kidney transplantation, UPCR: urine protein–creatinine ratio, ESRD: end-stage renal disease, FSGS: focal segmental glomerular sclerosis, ADPKD: Autosomal dominant polycystic kidney disease, GN: glomerulonephritis, MPGN: membranoproliferative glomerulonephritis, VUR: vesicoureteral reflux. * Statistical analysis using Mann–Whitney U test.

**Table 2 ijms-23-12253-t002:** miRNA expression pattern (CAD vs. Control).

Variate	miRNA Level (Mean Ct)				log2Ratio	*p* Value
	CAD	N	Control (n = 20)	N		
hsa-miR-122-5p	28.80 ± 1.19	19	27.34 ± 1.96	19	−1.45 ± 0.26	0.0097
hsa-miR-101-3p	26.87 ± 1.13	20	26.20 ± 0.66	20	−0.67 ± 0.18	0.0288
hsa-miR-15a-5p	28.09 ± 1.38	20	27.28 ± 1.60	20	−0.80 ± 0.47	0.0246
hsa-miR-20a-5p	28.73 ± 1.18	15	30.29 ± 1.05	7	1.55 ± 0.24	0.0132
hsa-miR-21-5p	26.92 ± 0.64	20	26.20 ± 0.42	20	−0.72 ± 0.62	0.0002
hsa-miR-589-5p	28.55 ± 0.41	9	29.68 ± 0.90	5	1.13 ± 1.27	0.0077
hsa-miR-29b-3p	29.42 ± 1.39	16	30.75 ± 1.72	12	1.32 ± 0.47	0.0329
hsa-miR-126-3p	27.89 ± 0.90	20	27.35 ± 0.59	20	−0.54 ± 0.14	0.0317

The expression signature of miRNA in CAD and control groups (CAD vs. control). Normalized Cq level fold change (log2) of ≥±0.585 and *p* value < 0.05 were considered significant. The qPCR data were analyzed according to M. Ahmed et al. [13].

**Table 3 ijms-23-12253-t003:** Top 10 Enrichment GO terms of biological process.

GO Term	Gene Ratio	Bg Ratio	*p* Value
Response to abiotic stimulus	108/386	441/2689	3.00 × 10^−10^
Extracellular matrix organization	49/386	142/2689	4.04 × 10^−10^
Extracellular structure organization	49/386	142/2689	4.04 × 10^−10^
External encapsulating structure organization	49/386	142/2689	4.04 × 10^−10^
Response to lipid	94/386	367/2689	4.87 × 10^−10^
Muscle cell proliferation	43/386	120/2689	1.39 × 10^−9^
Regulation of cellular component movement	105/386	440/2689	3.06 × 10^−9^
Tube development	112/386	482/2689	4.06 × 10^−9^
Negative regulation of signal transduction	109/386	477/2689	2.05 × 10^−8^
Neuron death	52/386	178/2689	8.26 × 10^−8^

The target genes set of the differentially expressed microRNAs were identified, and we found miRNA target interaction (MTI) in miRTarBase. These targeted genes were used as input data for enrichment analysis. GO stands for Gene Ontology database, and it was used to identify the top ten terms in the category of molecular function (MF), biological process (BP) and cellular component (CC) of the gene sets. GeneRatio was the ratio of gene number under the GO term to total gene number of total MTI genes. BgRatio was the ratio of total gene number under the GO term to total gene numbers in the whole database.

## Data Availability

The data presented in this study are available on request from the corresponding authors. The data are not publicly available due to ethical restrictions.

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
