# Peer review of "Association between Circulating MicroRNAs (miR-21-5p, miR-20a-5p, miR-29b-3p, miR-126-3p and miR-101-3p) and Chronic Allograft Dysfunction in Renal Transplant Recipients"

_ijms, 2022, doi:10.3390/ijms232012253_

Round 1

Reviewer 1 Report

The patient number is only 20.

Figure 1 Put the bar of the standard deviation.

“Grey bars indicated miR-20a5p, mir-122-5p and miR-589-5p were not normally distributed in both groups, thus Mann-Whitney U test were implemented. The other three miRNAs were showed in white bars, tested with Student’s T test.” Explain more the details.

Table 3: Title; Top 10 Enrichment GO terms of biological Process. What is for? Explain more for the relation between the MiR and the GO terms.  Please explain Gene ratio and Bg ratio.

Line87: miRTarbase. Please cite the references.

Figure 2: The authors should cite the reference.

The authors can discuss more regarding the roles an function of miRs.

Luo, W., Li, G., Yi, Z., Nie, Q., & Zhang, X. (2016). E2F1-miR-20a-5p/20b-5p auto-regulatory feedback loop involved in myoblast proliferation and differentiation. Scientific reports, 6, 27904. https://doi.org/10.1038/srep27904

Li, Z., Jiang, C., Ye, C., Zhu, S., Chen, X., Wu, W. K., & Qian, W. (2018). miR-10a-5p, miR-99a-5p and miR-21-5p are steroid-responsive circulating microRNAs. American journal of translational research, 10(5), 1490–1497.

Ghorbanmehr, N., Gharbi, S., Korsching, E., Tavallaei, M., Einollahi, B., & Mowla, S. J. (2019). miR-21-5p, miR-141-3p, and miR-205-5p levels in urine-promising biomarkers for the identification of prostate and bladder cancer. The Prostate, 79(1), 88–95. https://doi.org/10.1002/pros.23714

Zang, J., Maxwell, A. P., Simpson, D. A., & McKay, G. J. (2019). Differential Expression of Urinary Exosomal MicroRNAs miR-21-5p and miR-30b-5p in Individuals with Diabetic Kidney Disease. Scientific reports, 9(1), 10900. https://doi.org/10.1038/s41598-019-47504-x

Olivieri, F., Prattichizzo, F., Giuliani, A., Matacchione, G., Rippo, M. R., Sabbatinelli, J., & Bonafè, M. (2021). miR-21 and miR-146a: The microRNAs of inflammaging and age-related diseases. Ageing research reviews, 70, 101374. https://doi.org/10.1016/j.arr.2021.101374

Ye, M., Wang, S., Sun, P., & Qie, J. (2021). Integrated MicroRNA Expression Profile Reveals Dysregulated miR-20a-5p and miR-200a-3p in Liver Fibrosis. BioMed research international, 2021, 9583932. https://doi.org/10.1155/2021/9583932

Hromadnikova, I.; Kotlabova, K.; Krofta, L. Cardiovascular Disease-Associated MicroRNAs as Novel Biomarkers of First-Trimester Screening for Gestational Diabetes Mellitus in the Absence of Other Pregnancy-Related Complications. Int. J. Mol. Sci. 2022, 23, 10635. https://doi.org/10.3390/ijms231810635

The reviewer could not find Supplementary Materials. Please show them.

Reviewer 2 Report

the manuscript evaluates the expression of some miRNAs in chronic allograft dysfunction.

The study enrolled 40 kidney transplantation recipients with late graft survival for more than 17 10 years were enrolled.

The results are very interesting for the development of therapeutic markers.

I recommend changing the order of the paragraphs. The last paragraph should be discussion. Furthermore, the expression of miR-21-5p, miR-20a-5p and miR-101-3p should be investigated. In what other pathologies are they under-expressed or over-expressed?

It is advisable to mention these articles:

Doi: 10.3390 / diagnostics11010032

DOI: 10.3390 / diagnostics11010064

Round 2

Reviewer 1 Report

The manuscript became much better in form.

Materials and Methods

4. Patents?

should be "Patients", italic and not bold.

Author Response

Dear reviewers,

Thanks for your kind suggestion, we had correct the spelling mistake and style of the title. Some minor errors were corrected after examination again. The revision process really polish the article.